# Waterscapes for Promoting Mental Health in the General Population

**DOI:** 10.3390/ijerph182211792

**Published:** 2021-11-10

**Authors:** Xindi Zhang, Yixin Zhang, Jun Zhai, Yongfa Wu, Anyuan Mao

**Affiliations:** 1Department of Landscape Architecture, Gold Mantis School of Architecture, Soochow University, Suzhou 215123, China; 20205241005@stu.suda.edu.cn (X.Z.); info@eastscape.com (J.Z.); yuanke@yuankest.cn (A.M.); 2China-Portugal Belt and Road Cooperation Laboratory of Cultural Heritage Conservation Science, Research Center of Landscape Protection and Ecological Restoration, Soochow University, Suzhou 215123, China; wuyongfa@suda.edu.cn; 3Collaborative Innovation Center of Architecture and Urban Environment of Soochow University-Suzhou Yuanke, Soochow University, Suzhou 215123, China

**Keywords:** ecosystem services, abiotic elements, biodiversity, therapeutic landscapes, exercising

## Abstract

The WHO estimates that, with the development of urbanization, 25% of the population is suffering from psychological and mental distress. Preliminary evidence has suggested that aquatic environments and riparian areas, i.e., waterscapes, can benefit psychological and mental wellbeing. The aim of this study was to identify the processes of waterscape psychological and mental health promotion through aliterature review. We propose a design framework of waterscapes for achieving psychological and mental health in the general population that often visits waterscapes, which has the function of therapeutic landscapes through values of accessibility, versatility, habitats, and biodiversity. According to theories, waterscapes can improve psychological and mental health to divert negative emotions through mitigation (e.g., reduced urban heat island), instoration (e.g., physical activity and state of nature connectedness), and restoration (e.g., reduced anxiety/attentional fatigue). By accessing water (e.g., streams, rivers, lakes, wetlands, and the coast) and riparian areas, people can get in close contact with nature and spend more time in activities (e.g., walking, exploring, talking, and relaxing). Waterscapes with healing effects can enhance psychological resilience to promote people’s psychological and mental health. Future research should focus on ensuring an adequate supply of waterscapes and promoting the efficiency of waterscape ecosystem services on mental health. Moreover, fora deep understanding of the complexity of nature–human health associations, it is necessary to explore more consistent evidence for therapeutic waterscapes considering the characteristics and functional mechanisms of waterscape quality, in terms of freshness, luminescence, rippling or fluidity, and cultural value, to benefit public health and biodiversity conservation.

## 1. Introduction

The World Health Organization (WHO) defined human health as ‘a state of complete physical, mental, and social wellbeing and not merely the absence of disease or infirmity’ (WHO, 1948), and the definition has not been amended since then. However, about 29% of adults worldwide are experiencing common mental distress, including depression [1]. With rapid urbanization, by 2050, about 68% of the population in the world will reside in urban areas, which will further increase this patient number. The rising incidence of psychological and mental distress necessitates the search for an effective solution to improve mental health.

Evidence has shown that natural spaces, especially urban green spaces and waterscapes (water and riparian areas), can lower exposure to air pollution, extreme heat, and noise [2,3], increase opportunities for physical activity and social engagement, and decrease stress and depression by contacting with nature [4,5,6]. Considering these services, urban natural spaces are great potential buffers to protect and promote human health [7,8], including physical and mental health [9,10,11,12,13]. This positive relationship between natural spaces and human health can be explained by three approaches: (a) the Attention Restoration Theory (ART), (b) the Psycho-physiological Stress Recovery Theory (PSRT), and (c) the Biophilia Hypothesis [14]. Briefly, ART suggests that restorative environment can alleviate mental fatigue caused by extensive episodes of concentration and focus [15], and PSRT believes that humans will have a direct, instinctive positive emotional response to nature that affects both psychological and physiological measures as they evolve in their natural environment [16]. Additionally, the Biophilia Hypothesis suggests that human beings have preferred natural environments and evolved an innate emotional connection to other living organisms [17] (pp. 31–41), [18].

Water as the most important physical and aesthetic landscape element is crucial in multiple ecosystem services, linking environmental psychology, landscape design, and social and culture research [3,19]. However, the literature about the relationship between water and human health has been more extensively assessed in environmental toxicology and microbiology in the past, and it mainly focused on drinking water security [20,21]. Recently, attention has turned to waterscapes and their benefits toward human wellbeing [22,23]. Waterscapes are landscapes with an expanse of water which consist of water bodies and associated riparian habitats, including streams, rivers, lakes, wetlands, the coast, and their riparian zones, which can service psychological health benefits via the mechanisms of mitigation, instoration, and restoration [24,25]. Studies have indicated that waterscapes can play an important therapeutic role in the physical, mental, and social health of residents, e.g., by lowering psychological distress [26,27,28,29]. Additionally, the literature suggests that a neighborhood waterscape is more beneficial for psychological and mental health than a neighborhood green space [30,31].

These areas with physiological and psychological therapeutic benefits can be called the ‘therapeutic landscape’ [32,33,34]. With the increasing pressure of life, the demand for medical resources, such as psychological support and relief, is increasing. Through therapeutic benefits of waterscapes, the stress of medical treatment can be relieved. However, the benefits of ‘waterscapes’ for areas with water areas (e.g., lakes, rivers, streams, and wetlands) and their riparian areas are less studied compared to green space [35,36], and the results of some studies have also been mixed [37]; thus, the positive associations of waterscapes with mental health and their functional mechanisms deserve more systematic attention [38].

A biennial nature–human relationship is crucial for promoting public mental health [39,40,41]. Waterscapes have been shown to be health resources for psychological and mental health promotion. However, more research attention has beenpaid to green space and its psychological and mental health benefits. In order to fill the gap of limited research in the psychological and mental health promotions of waterscapes and their mechanisms, this article focuses on waterscapes for mental health of the general population (who often visit waterscapes); accordingly, the objectives of this study were (a) to identify the processes of ‘waterscapes’ for psychological health promotions, including biotic elements, abiotic elements, biodiversity, and exercising, and (b) to propose the design principles of waterscapes which play the role of therapeutic landscapes in terms ofimproving accessibility, increasing interactivity, and restoring water quality and biodiversity.

## 2. Methods

### 2.1. Search Strategy

Electronic journal databases (Web of Science, Science Direct, Google Scholar, WHO, and other databases) were used to collect over 250,000 studies in the last 20 years by searching the topic words separately at first, including psychological health (set 1: ‘psychological health’ or ‘mental health’), waterscapes (set 2: ‘blue space’, ‘water’, or ‘waterside’), and exercise (set 3: ‘exercising’, ‘walking’ or ‘swimming’). Then, psychological or mental health was combined with keywords such asblue space, water, waterside, and exercise for another search. Furthermore, we collected and reviewed references in relevant articles and some articles with high impact outside of the last 20 years, such as Herzog (1985) [42], Ulrich et al. (1991) [16], Gesler (1992) [43], Wilson (1993) [17], and Kaplan (1995) [15]. After excluding duplicates and irrelevant studies, we obtained 112 studies for our review (Figure 1). To be included in the paper, the literature had to meet the following criteria: (a) first-hand research articles and articles focusing on water and riparian areas were preferred, (b) findings were directly applicable to mental health promotion in waterscapes, and (c) articles had high impact.

Finally, after collecting the literature, the articles were mainly divided into two kinds which are shown in Table 1: (a) the therapeutic benefits of waterscapes on psychological health (set 1, set 2), and (b) the positive influence of exercising in waterscapes on psychological health (set 1, set 3). Through reading and analyzing, these articles were listed in categories to be summarized.

### 2.2. Research of Design Principles and Strategies

Publications about accessibility, versatility, habitats, and biodiversity have indicated that water and riparian areas can benefit mental health [62]. Both natural and built scenes with water are associated with higher preferences, greater positive impacts, and higher perceived resilience than scenes without water [68]. The public values access to waterscapes for relaxation, as well asactivities of blue recreation or social connections, to avoid delaying or losing recreational and interpersonal experiences and associated health benefits [73,75].

In addition, the objectives of this article were (1) to summarize the design principles of health-friendly waterscapes for psychological benefits, including improving accessibility, increasing interactivity, and restoring water quality and biodiversity by analyzing the mechanisms of waterscapes in promoting mental health, and (2) to propose the conceptual design strategies of building a ‘therapeutic landscape’.

## 3. The Concept of ‘Therapeutic Landscapes’

Initially, ‘therapeutic landscapes’, a series of areas which not only promote physical health, but also purify and soothe the mind and regulate poor psychological conditions, were mainly thought to consist of places such aspilgrimage sites and spas [43]. The power of faith led people to believe that the water in these places could bring healing [82,83]. With the development of psychological healing research, more researchers paid attention to some spaces with potential therapeutic abilities, such as parks, gardens, woodlands, riverside water spaces, lakes, and beaches [33,55]. Therefore, the meaning of ‘therapeutic landscapes’ has grown to encompass the benefits of health promotions, including building capacities, strengthening physical fitness, releasing stress, and relaxing [46,80].

The ‘therapeutic landscape’ in this paper mainly refers towaterscapes, i.e., streams, rivers, lakes, wetlands, and their riparian areas, which are landscapes with water at the center of the environment with therapeutic functions for mental health [49,56,65]. In these spaces, the ‘cleansing’ or ‘purifying’ abilities of water can also mediate emotions and calm visitors, thus benefiting mental health [32].

## 4. The Psychological Benefits of ‘Therapeutic Landscapes’: Waterscapes with Eco-Healing Functions

An investigation in the UK showed that people living near the coast tend to be healthier than those living inland [67]. The restoration experience is stronger in a riverside environment than in an urban built environment or urban green space (mainly parks) [58]. Natural environments have great potential for health benefits and the wellbeing of entire populations [63,78]. Considering the above, a framework of the psychological and mental benefits from (a) waterscapes, and (b) physical exercising in waterscapes is proposed (Figure 2).

### 4.1. The Psychological Benefits from Waterscapes

An immersive therapeutic experience has indicated that waterscapes can not only give people benefits in the material aspect, but also reach the height of cognitive and emotional release [32,45,55], which is related to ART [15]. Landscapes with water, which are called as waterscapes, have proven to be more attractive than other areas without water [84], and they are also better for promoting psychological health [52]. The sounds of running water can also promote psychological and mental health by reducing noise and enhancing urban soundscape [57]. Some evidence has also pointed to the effectiveness of waterscapes in the promotion of psychological benefits as related to biodiversity, space quality, and location form [14,42,85].

#### 4.1.1. Biotic Elements

It has been proved that bird and fish watching can not only lower pulse rate and muscle tone and increase skin temperature, but also bring about greater benefits for psychological health [48], which is related to the Biophilia Hypothesis [17]. A survey among 100 adult university students taken in a quiet room in the School of Psychology, Queen’s University Belfast showed that viewing certain animals, such as fish and birds, could reduce the cardiovascular response to psychological stress and help alleviate anxiety [86].

#### 4.1.2. Abiotic Elements

a. Water features—The water features of sounds, color and clarity are important sensory perceptions of waterscapes [65,85]. This can be due to people appreciating the sounds of water [69], especially the diversity and specificity of these sounds, from the calm cascade to the vibrant roar [68]. The sound levels of waterscapes can be divided into four levels: 58–72 dB (fountain), 64–75 dB (waterfall), 48–61 dB (pond), and 52–68 dB (stream). It has been proved that water sounds with low decibels are more popular than water sounds with high decibels and large frequency amplitudes, whether low frequency (<500 Hz) or high frequency (>3000 Hz) [87,88]. It has also been shown that noise reduction and enhancement of urban soundscapes are two important ways in which nature’s sounds promote psychological and mental health [57,86].

b. Accessibility—Accessibility, as a factor influencing people’s frequency of using waterscapes, indirectly affects the contribution of waterscapes to psychological health [55]. It has been indicated that a <15 min walk and a 300 m distance to water bodies are reasonable indicators for accessibility [89]. A study focused on urban neighborhoods in Northern Utah indicated that, for every 10 m increase in distance from an access point, residents were less likely to stay in waterscapes. In addition, households living immediately adjacent to a riverside were more likely to spend time there and more likely to be positively influenced by playing on the riverside and experiencing its sights and sounds [55], which is related to PSRT [16].

#### 4.1.3. Biodiversity

Biodiversity within cities is critical to human health and wellbeing and provides a wide range of important ecosystem services [90,91,92]. Four domains of pathways link biodiversity with health, which are both beneficial and harmful, including reducing harm (e.g., decreasing exposure noise pollution), restoring capacities (e.g., attention restoration, stress reduction), building capacities (e.g., promoting physical activity), and causing harm (e.g., dangerous wildlife, zoonotic diseases, and allergens) [93]. Biodiversity observation can have an effect on increasing regular physical exercise which can encourage greater appreciation of the environment and benefit psychological and emotional health as more time is spent in natural settings [70,90]. The reduction in positive human–nature interaction is caused by a combination of biodiversity loss coupled with the growth of sedentary pastimes and perceived safety issues that limit activities [71]. Cracknell et al. (2017) [14] conducted a study among 79 participants in UK and investigated the effect of marine biodiversity on psychological wellbeing in a marine aquarium setting, and they found that higher biodiversity will result in better psychological health. In contrast, evidence has also indicated that the benefits of real nature will be even greater than simulated nature through determining whether a simulated environment can represent a real environment [94]. Accordingly, in addition to the two established aspects of blue environments, i.e., water features and accessibility, biodiversity makes a significant contribution and is inextricably linked to psychological health [90,95].

### 4.2. The Psychological Benefits Coming from Exercising in Waterscapes

The waterside areas of urban blue spaces are particularly stimulating for dynamic activities, and it is indicated that living closer to waterscapes will be associated with higher levels of physical activity [54,74,96], such as walking, exploring, talking, and relaxing, which has the same positive impact on people’s mental health as being active in urban green spaces [49]. Additionally, several respondents living within four neighborhoods in two coastal towns mentioned their preference for three types of social dynamics, each of which was evident in waterscapes: (a) seeking friendly conversation and a joyful atmosphere, (b) engaging in spaces that offering a variety of opportunities for family leisure and wellbeing, and (c) bonding relationship through shared hobbies and experiences [32]. A program in Europe, named the ‘Blue Gym’, tries to encourage people to join in exercises in coastal areas to improve psychological health, such as swimming, sailing, surfing, walking, and rambling [47].

## 5. The Design of Psychological Health-Friendly Waterscapes

It has been shown that riverside districts of urban blue spaces are potential areas in the promotion of psychological health. Those who visit waterscapes regularly are more likely to have good psychological health [44], and those who can see waterscapes from their residence are more likely to report good general health. People are also more likely to visit waterscapes if they feel there are good facilities and wildlife to see [50]. As such, it is necessary to think about how to make the best use of these areas and make them a psychological health-friendly landscape [97,98].

### 5.1. The Design Principles of Psychological Health-Friendly Waterscapes

Psychological health-friendly waterscapes are mainly designed for human viewing enjoyment and a sense of participation. Therefore, designers should create a landscape which is beautiful, comfortable, and participatory [55]. In conclusion, the design principles of waterscapes can be summarized into (a) accessibility (set 1), (b) versatility (set 2), and (c) habitats and biodiversity (set 3).

Set 1: AccessibilityImprove accessibility to the riverside or riparian zone to increase frequency of use.Set 2: VersatilityIncrease the opportunities of people getting into contact with nature;Increase recreational spaces in the riverside areas so that people can spend more time there, which includes increasing the versatility of water features to increase ornamental appeal and increasing the height variation of the river to enrich the sound from it.Set 3: Habitats and biodiversityEnhance the self-purification capacity of rivers to maintain water quality;Increase habitat quality to improve river ecosystem functioning;Increase habitat heterogeneity and complexity to increase biodiversity.

### 5.2. The Design of Psychological Health-Friendly Waterscapes

With the development of urbanization, urban waterscapes, including streams, rivers, lakes, and their riparian areas, have been impacted by poor water quality, low aquatic biodiversity, low connectivity, and less accessibility. Therefore, how to renew the site vitality and update riversides to health-friendly waterscapes needs more discussion. For this, the conceptual design strategiesof building urban streams and rivers to be psychological health-friendly waterscapes are introduced below.

#### 5.2.1. Site Analysis

As a result of urbanization, the population of cities is increasing. Furthermore, this phenomenon has led to increasing tension in urban land, resulting in the gradual conversion of many natural ecological sites into building sites. By investigating and analyzing the current situation of Suzhou urban rivers in December 2019, we found that urban rivers in Suzhou are mostly blocked off from urban riverfront neighborhoods by high-fence walls because of current problems such as poor river water quality. Although these high-fence walls block the negative views of the river, they also block the opportunities for residents to take relaxing walks along the river and cut off the communication between residents and the river. In many urban areas, in order to enhance the utilization of riverside land, river revetments are also dominated by vertical revetments, which block the connectivity of the material and energy exchange between water bodies and land. As discussed above, four conceptual therapeutic landscape dimensions, i.e., experiential, symbolic, social and physical activity space [65], are out of service, suchthat the psychological health promotions of urban rivers will not be well served for enhanced contemplation, emotional connection, and participation.

In order to resolve the conflict between the current problems of the river and the needs of the residents, the conceptual design strategies for a psychological health-friendly landscape are discussed next.

#### 5.2.2. Site Design

As discussed above, waterscapes can promote psychological health through (a) getting in touch with nature (e.g., visual contact with animals and listening to sounds from nature), (b) providing spaces for exercising (e.g., walking, talking, and relaxing), and (c) creating biodiversity-rich sites [99]. To meet these three main pathways by following design strategies, the site conceptual design shouldfocus on (a) improving accessibility, (b) increasing interactivity, and (c) restoring water quality and biodiversity.

a.Improving accessibility

An increase in distance will result in a reduction in access to waterscapes, which will then reduce the psychological health benefits of waterscapes [79]. High-fence walls along the river block residents’ access to the riverside, thus reducing the promotion benefits of waterscapes for psychological health. Therefore, in order for waterscapes to play a role in psychological health, the first step is to improve accessibility to residents [100,101] by (a) removing parts of the unnecessary high-fence walls, (b) designing alleyways to increase the opportunities for residents’ access to waterscapes, or (c) designing trestles to strengthen the connection between two sides of the river (Figure 3).

b.Increasing interactivity

In summary, two important pathways of waterscapes in promoting psychological health are getting in touch with nature and providing spaces for people to relax, including viewing animals (e.g., birds and fish) [48], listening to the sounds of nature (e.g., birdsong and insects chirping) [86], exercising in waterscapes (e.g., walking, running, and relaxing) [49], nature education in schools, and other outdoor water-based activities related to active citizenship and environmental awareness [72]. For these, we need to design spaces conducive to activities, water-friendly platforms, leisure facilities, etc. (Figure 3).

c.Restoring water quality

According to a review of the literature, water quality can influence the benefits of waterscapes on psychological health; thus, mechanisms to restore them for psychological promotion must be considered [90,95].

Water remediation techniques mainly include physical remediation and bioremediation. Bioremediation for improving water quality, including microbial remediation, phytoremediation, and animal remediation, is a biological process that uses specific organisms to eliminate or enrich environmental pollutants under certain conditions, thereby restoring the polluted water [102].

Experimental results have shown that the removal efficiency of nitrogen in water increases with the increase in plants bed coverage [103], where a coverage of 5–38% represents the best setting [104]. In addition, adding rice straw and light ceramsite to plant beds forming two different constructed wetlands can achievethe bioremediation of eutrophic water under low-temperature conditions [105]. Some experiences have indicated that plant technologies whereroots are in permanent contact with water can remove pollutants through biological processes, and their benefits are not only water quality restoration, but also landscape protection and aesthetic benefits [106,107].

Additionally, microbial remediation mainly uses the metabolic function of microorganisms to remove organic matter from water. Animal remediation techniques, including the placement of fish, daphnia, snails, and mussels, can inhibit the overgrowth of algae in water and alleviate the problem of nitrogen and phosphorus pollution [108]. In practice, silver carp, bighead carp, and other fish which feed mainly on phytoplankton are more reliable choices [108].

d.Restoring biodiversity

Biodiversity can enhance the effect of waterscapes on psychological health [97,98]. The restoration of biodiversity is primarily about creating urban areas that provide onsite benefits to native species and ecosystems by providing essential habitats and food resources [92], which will be promoted by plant restoration [109]. Biodiversity can be restored and protected through restoration of ecological processes. In addition, restoring the urban environment will help change negative human behaviors toward nature through exercise and provide a place to experience nature [110,111].

Therefore, some solutions, such as ecological floating islands, ecological revetments, and ecological planting ponds, can promote water quality purification and habitat creation to enhance complexity and heterogeneity, support biodiversity restoration, and promote riparian water-based activities (Figure 3). These spaces and habitats provide conditions for water–land material, nutrient and organism exchange, plant growth, and aquatic animal life, while contributing to the psychological health-promoting effects of waterscapes.

### 5.3. Practice and Future Research

Evidence has shown that waterscapes can improve psychological and mental health [51], including when exercising in them. However, many studies only investigated the mechanisms of promoting psychological and mental health promotion, and there are few studies focusing on the design of psychological health-friendly waterscapes. In order to promote the research on the psychological and mental health benefits of waterscapes, we need to combine theories with practice. Accordingly, landscape planners and designers should explore the design means of psychological health-friendly waterscapes, and then apply the theories to landscape design to meet the social needs.

## 6. Conclusions

Due to rapid urbanization, the increasing pressure of life has led to an increase in pressure on people’s minds, which can cause mental distress including depression; therefore, the need of psychological support is growing. It has been shown that waterscapescan be a therapeutic landscape [76], and it has also been indicated that waterscapes can have more benefits than green spaces in psychological and mental health [30]. The water features, accessibility, and biodiversity are the potential key factors thataffect psychological health promotion benefits. Future research should focus more on investigating the efficiency of the promoting effect of waterscapes on psychological health, including the design of psychological health-friendly waterscapes.

### 6.1. The Benefits of Waterscapes on Psychological Health Promotion

The benefits of waterscapes on psychological health can be seen in two aspects: (a) the contribution of the waterscape itself to psychological health, and (b) exercising in waterscapes. It is also indicated that waterscapes can improve psychological health through mitigation (e.g., reduced urban heat island), instoration (e.g., physical activity and state nature connectedness), and restoration (reduced anxiety/attention fatigue) [24]. Moreover, biotic elements, abiotic elements, and biodiversity are potential factors that affect the promotion benefits of a waterscape as a ‘therapeutic landscape’.

### 6.2. The Design of Psychological Health-Friendly Waterscapes as a Therapeutic Landscape

For enhancing the psychological benefits of waterscapes, we need to consider approaches for how to renew urban waterscapes and improve their efficiency to support psychological health. By exploring the mechanisms of waterscapes on promoting psychological health, including biotic elements, abiotic elements (e.g., water features and accessibility), and biodiversity, the design principles can be summarized into three major aspects: (a) improving accessibility, (b) increasing interactivity, and (c) restoring water quality and conserving biodiversity.

### 6.3. Practice andFuture Direction of Research in ‘Waterscapes’ as a Therapeutic Landscape

Waterscapes are an important resource for wellbeing and health, while benefitting psychological and mental fitness [35,51] with cultural ecosystems services [112]. Further research is needed to provide more consistent evidence and detailed information for the use of therapeutic waterscapes considering the characteristics and functional mechanisms of waterscape quality, in terms of freshness, luminescence, rippling or fluidity, and cultural value, to promote public mental health and biodiversity conservation. In addition, when exploring complexity and dynamics in nature–health associations, there is a need to clarify the effectiveness of each psychological health-promoting factor to help people better design nature-based psychological health-friendly waterscapes.

## Figures and Tables

**Figure 1 ijerph-18-11792-f001:**
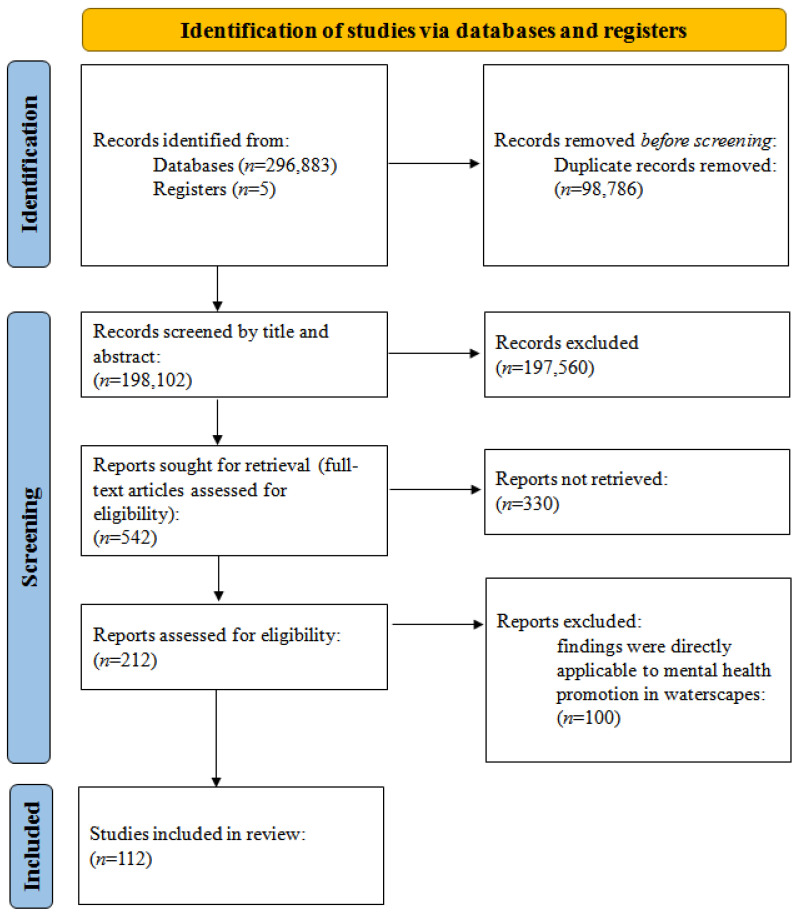
Preferred Reporting Items for Systematic Reviews and Meta-Analyses (PRISMA) flow diagram showing the numbers of studies screened and includedin the article.

**Figure 2 ijerph-18-11792-f002:**
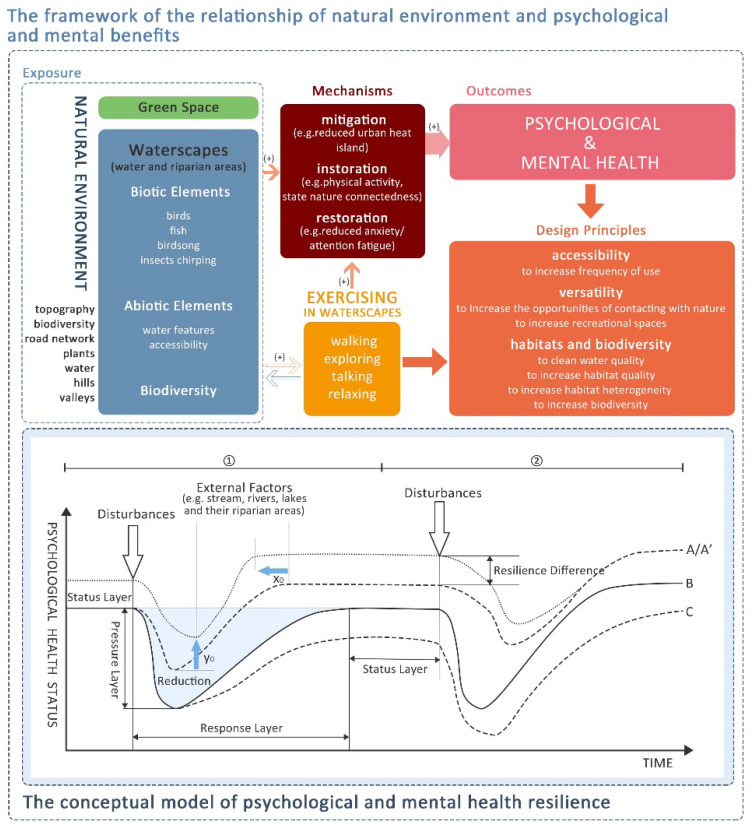
The framework of the relationship between the natural environment and psychological and mental benefits consists of two parts. The first half demonstrates the process of the natural environment (including exercising in it) for psychological and mental health, as well as shows the mechanisms between them. In addition to this, the design principles of psychological health-friendly waterscapes are included. The second half is the conceptual model of psychological and mental health resilience. The model shows the recovery process of psychological health after a disturbance which includes negative factors such as stressful events, fears, insecurities, and anxiety. After disturbance ①, psychological health status loses its balance to reduce its psychological health level. Through resilience capacity, it then returns to the original state (e.g., B). Then, after disturbance ②, psychological health status is reduced and reaches a new state (e.g., A, C, and ② of B). A, B, and C represent the psychological health resilience from high to medium to low. A and A’ also show that the same or different equilibrium states may exist in psychological health at the same conditions. x_0_ denotes the difference in response time (time of return to normal or better status), and y_0_ denotes the difference in psychological health resilience (the lowest psychological health status). The differences in response status (y_0_) and time (x_0_) are caused by external factors which are linked to different kinds of external environments, such as green, water, and other natural environments. In this paper, the external factor constitutes waterscapes such as streams, rivers, lakes, and their riparian areas. It means that people living near to or with access to waterscapes will have higher psychological health resilience and a shorter response time.

**Figure 3 ijerph-18-11792-f003:**
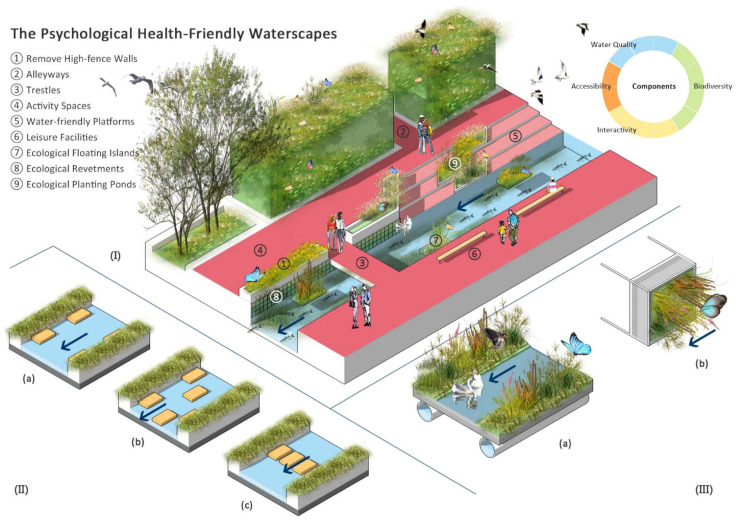
(**I**) Strategies for improving accessibility, increasing interactivity, and restoring water quality and biodiversity. Removing high-fence walls and designing alleyways and trestles allow improving accessibility. Designing activity spaces, water-friendly platforms, and leisure facilities allow increasing interactivity. Lastly, designing ecological floating islands, ecological revetments, and ecological planting ponds allow restoring water quality and biodiversity. The pie chart in the upper right corner shows the importance of each component in psychological health-friendly waterscapes (from high to low: biodiversity > interactivity and water quality > accessibility. (**II**) Different arrangements of ecological floating islands at different water velocities: (**a**) when water velocity is high, floating islands are placed along revetments to open the water channel; (**b**) when water velocity is medium, floating islands are placed staggered to slow down velocity and increase the contact time between the water and plants; (**c**) when water velocity is slow, floating islands are placed side by side and vertical to revetments to achieve water purification. (**III**) Structure of (**a**) ecological floating islands and (**b**) ecological revetments.

**Table 1 ijerph-18-11792-t001:** The table shows some of the literature discussing (a) the therapeutic benefits of waterscapes on psychological health, and (b) the positive influence of exercising in waterscapes on psychological health.

Study Characteristics	
Aspect	Study
(a) the therapeutic benefits of waterscapes on psychological health	[10,14,15,16,23,24,25,26,27,30,31,32,33,35,37,38,39,40,42,44,45,46,47,48,49,50,51,52,53,54,55,56,57,58,59,60,61,62,63,64,65,66,67,68,69]
(b) the positive influence of exercising in waterscapes on psychological health	[4,6,9,29,32,47,49,54,70,71,72,73,74,75,76,77,78,79,80,81]

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
