# Peer review of "Waterscapes for Promoting Mental Health in the General Population"

_ijerph, 2021, doi:10.3390/ijerph182211792_

Round 1

Reviewer 1 Report

Thank you for the effort to improve the manuscript and congratulations on the achievement.

Author Response

Thank you very much for your recognition of our past work and for your suggestions. And we also appreciate your positive comment.

Reviewer 2 Report

Please find in attachment.

Author Response

We sincerely thank reviewer for the valuable comments, which are of great help in revising the manuscript. Accordingly, the manuscript revised with comments has been systematically improved. Our responses are following. The page and line number mentioned below are in accordance with the number in revised manuscript.

Responses to comments

the paper is more suitable as review than a systematic review.

Response: After re-thinking, we consider that our manuscript is more suitable as a review. And we have modified the title of the manuscript as 'Waterscapes for Promoting Mental Health of General Population - A Review'.

the title should indicate that this paper is a review and population mental health of adults? general population? who?

Response:The paper is a review and population mental health of general population. So we have modified the title of the manuscript as 'Waterscapes for Promoting Mental Health of General Population - A Review'.

give full terms for who.

Response:We thank reviewer for pointing this out. It has been revised as 'We propose a design framework of waterscapes for achieving psychological and mental health of general population who often visit waterscapes, which has the functions of therapeutic landscapes through values of accessibility, versatility and habitats and biodiversity' (Tracked version, Page 1; L16-17), and ' In order to fill the gap of limited research in psychological and mental health promotions of waterscapes and their mechanisms, the article focuses on waterscapes for mental health of general population (who often visit waterscapes) and the ob-jectives of this study are:' (Tracked version, page 2; L98-102)

psychological and mental distress please be consistent sometimes distress sometimes problems its really confusing. this paper should focus mainly on quality of life. giveevidence from case studies.

Response:We thank reviewer for pointing this out. It has been unified as 'psychological distress'.

remove 350 mil people with depression this paper has nothing to people with depression.

Response:We thank reviewer for pointing this out. It has been removed.

waterscape is need clear definition as a landscape in which an expanse of water is a dominant feature.

Response:We thank reviewer for pointing this out. It has been revised as 'Waterscapes are landscapes with an expanse of water which consist of water bodies and associated riparian habitats' (Tracked version, Page 2; L72-75)

types of waterscapes need to be described.

Response:We thank reviewer for pointing this out. It has been described as 'Waterscapes are landscapes with an expanse of water which consist of water bodies and associated riparian habitats, including streams, rivers, lakes, wetlands, the coast, and their riparian zones, which can service psychological health benefits by the mechanisms of mitigation, instoration and restoration (White et al., 2020; White et al., 2021)' (Tracked version, Page 2; L73-77)

at the end of the introduction please describe the rationale for the review in the context ofexisting knowledge. it’s not clear what are the main questions of the current review.provide an explicit statement of the objectives or questions the review addresses.

Response:We thank reviewer for pointing this out. It has been revised as 'Waterscapes have been shown to be health resource for psychological and mental health promotion. However, more research attention is paid to green space and its psychological and mental health benefits' (Tracked version, Page 2; L95-98)

an emphasis needs to be here that this is a review not a systematic review. usually, reviewsdo not contain a PRISMA chart.

Response:We thank reviewer for pointing this out. We mainly use the PRISMA chart to display the collecting and screening process of literature. We think that the PRISMA chart can make the process to be clearer.

2.2 read as design research is this correct or it should be research design.

Response: Thanks reviewer for pointing this out. It has been revised as 'Research of design principles and strategies' (Tracked version, page4; L133)

figures are is not labeled. please provide clear footnote.

Response:Figures 1-3 have been numbered and illustrated, and indexed in the corresponding sections of the article.

cite each included study and present its characteristics. can you give descriptions wherethey were done and when and other details? the 112 studies included in the review.

it’s very important in the methods that authors describe how they reached to markersmeasurement e.g. some research reported that waterfall promotes vigorous feelings morethan the other waterscapes and they provided markers using profile of mood states (poms)questionnaire and a semantic differential (sd) scale we need to understand what each studyhas measured and why its included in this review.

Response:Thanks reviewer for pointing this out. Table 2 describes the main research perspectives contained in the literature, including (a) the therapeutic benefits of waterscapes on psychological health, and (b) the positive influence of exercising in waterscapes on psychological health. The article summarizes the conclusions of related publications on the promotion of psychological and mental health of waterscapes to search the mechanisms of waterscapes psychological and mental health promotion, and then we consider the design principles and strategies of psychological health-friendly waterscapes based on the current situation of Chinese cities. Therefore, the specific details of each study were not described and presented.

the sound levels of the four waterscape facilities should be discussed 58~72 db (fountain),64~75 db (waterfall), 48~61 db (pond), and 52~68 db (stream); this is important becausesound levels can influence the psychological impact.

Response:We thank reviewer for pointing this out. It has been revised as 'And the sound levels of waterscapes can be divided into four levels: 58-72 dB (fountain), 64-75 dB (waterfall), 48-61 dB (pond), and 52-68 dB (stream). It has been proved that water sound with low decibel is more popular than water sound with high decibel and large frequency amplitude, whether low frequency (<500 Hz) or high frequency (>3000 Hz) (Daria et al., 2020; Tedja&Tsaih, 2015).' (Tracked version, page 6; L215-219)

discussion – not clear what was the strengths and weaknesses of each of the includedstudies. please provide detailed examination and analysis. provide a general interpretationof the results in the context of other evidence. discuss any limitations of theevidenceincluded in the review, these are satisfaction studies mainly not clinical. discuss anylimitations of the review processes used for example you provided review and not meta-analysis. discuss implications of the results for practice, policy, and future research.

Response:We thank reviewer for pointing this out. Thearticle not only focuses on the literature review, but also discusses the design principles and strategies of psychological health-friendly waterscapes based on the conclusions of the relevant literature. So, we didn’t show too much details of each of the included studies.And the implications of the results for practice, policy, and future research have been revised as '5.3. Practice and future research: Many evidences have shown that waterscapes can improve psychological and mental health (Gascon et al., 2015) through themselves and exercising in them. However, many studies only stay at the research of the mechanisms of waterscapes psychological and mental health promotion at present, and there are few studies searching on the design of psychological health-friendly waterscapes. In order to promote the research on psychological and mental health promotion benefits of waterscapes, we need to combine theories with practice. Based on the site, landscape planners and designers should explore the design means of psychological health-friendly waterscapes, and then apply the theories to landscape design to meet the social needs.' (Tracked version, page 10; L403-412)

conclusion is review findings suggests that urban landscape planners need to considerwaterscape facility types when they plan urban green space to ensure that they createpsychologically comfortable places for users.

Response:Conclusion puts forward the suggestion that urban landscape planners should pay attention to waterscape design on the basis of summarizing the benefit of waterscape psychological and mental health promotion and the design principles and strategies of psychological health-friendly waterscapes.

declare any competing interests of review authors

Response:We thank reviewer for pointing this out. It has been revised as 'Declaration of interests: Authors declare that they have no known competing financial interests or personal relationships that could have appeared to influence the views of this paper.' (Page 11, lines 452-453)

Round 2

Reviewer 2 Report

the revision was prepared fast, but fairly effective. i commend authors. please work with production team of ijerph to provide better quality figures for final set. 

This manuscript is a resubmission of an earlier submission. The following is a list of the peer review reports and author responses from that submission.

Round 1

Reviewer 1 Report

This review summarizes the literature on the health and well-being benefits of urban blue space and water bodies. The authors combine the literature overview with a case study from Suzhou, China and propose some design ideas how the benefits of blue space can be improved. The ideas focus on improving accessibility, the possible uses of blue space (e.g. physical activity), water quality and biodiversity. Overall, this article is interesting to read but it may need major revisions. First, the English language of the manuscript needs to be improved. Second, the literature selection process needs to be explained in more details. However, the greatest issue might be related to the lack of novelty in this article. Over the past years, several review studies have been published on similar topics and research questions. Therefore, most of the content and conclusions of this new article do not seem new. The only potential unique aspect of this article may be the case study. In the following, I will list my issues and concerns in more detail.

Major issues:

Language:

  • It is quite evident that the overall quality of the written language needs some improvement. Some paragraphs contain sentences with inaccurate wording or grammar mistakes. I would recommend to ask an English native speaker to do some proofreading and rewriting.

Novelty:

  • Over the past years, several review studies have been published on similar topics and research questions (Gascon et al 2015, Völker and Kistemann 2011, Foley and Kistemann 2015, White et al 2020, McDougall et al 2020). Therefore, most of the content and conclusions of this new article do not seem very new. The only new aspect of this article is the unique case study from the city of Suzhou. However, this particular section is quite short and only describes very general ideas and concepts on how to improve blue space access, quality and biodiversity. What might be missing are technical details of the construction, descriptions of the project implementation and an analysis of the overall impact these changes had on the local residents. Adding these aspects to the manuscript would greatly improve the article.

Minor issues:

Page 2, L. 4-5: This section is a bit out of place and, to me, does not follow the logic of the paragraph.

Page 2, L. 9-10: I am missing a reference to this statement or a least an examples of review studies in the area of toxicology and microbiology. In addition, I am not sure weather research on blue space and human health has been so much neglected, since there have been several reviews published on this topic (Gascon et al 2015, Völker and Kistemann 2011, Foley and Kistemann 2015, White et al 2020, McDougall et al 2020).

Page 2, L. 41-50: This Methodology of the review is very short and does not explain in detail how the final list of 85 articles was selected. I would recommend that the authors use the PRISMA guidelines or any other established guideline for systematic reviews (Moher et al 2009).

Page 3, Figure 1: This figure seems a bit out a place. It nicely illustrates the human mental health recovery process after a disturbance, but the link to the health benefits of blue space is not very clear. Maybe the authors could add an explanation to the figure and its description.

Page 3, L. 9-10: The second objective is quite specific for a general literature overview on the design of natural and artificial blue space and blue space access. Could the authors please explain why you focus on the city of Suzhou. In general, could you please add this particular research aim to the introduction?

Page 4 Figure 2: Could the author please explain the novel aspects of their framework. In my opinion, the proposed framework is very similar to the framework published by (White et al 2020) and does not have any noticeable new aspects.

Page 5 L. 39-41 This sentence is hard to understand. Could you please rephrase it?

Page 5 L. 42-43 The study by Cracknell et al. (2016) investigates the effect of marine biodiversity on people’s psychological well-being in a marine aquarium setting. I would suggest to add this information to the paragraph and discuss the possible implications, since experienced biodiversity in zoos or aquariums is not comparable to experienced biodiversity in nature.

Page L. 51: Could you please add information on the “respondents”. Are they part of the study conducted by Bell et al. (2015)?

Page 6 L 49: Please write “relaxing walks” instead of just “relaxing”.

Page 7 L. 10: “As discussed above, …”

References:

Foley R and Kistemann T 2015 Blue space geographies: Enabling health in place Heal. Place 35 157–65 Online: http://dx.doi.org/10.1016/j.healthplace.2015.07.003

Gascon M, Mas M T, Martínez D, Dadvand P, Forns J, Plasència A and Nieuwenhuijsen M J 2015 Mental health benefits of long-term exposure to residential green and blue spaces: a systematic review Int. J. Environ. Res. Public Health 12 4354–79

McDougall C W, Quilliam R S, Hanley N and Oliver D M 2020 Freshwater blue space and population health: An emerging research agenda Sci. Total Environ. 737 140196 Online: https://doi.org/10.1016/j.scitotenv.2020.140196

Moher D, Liberati A, Tetzlaff J, Altman D G, Altman D, Antes G, Atkins D, Barbour V, Barrowman N, Berlin J A, Clark J, Clarke M, Cook D, D’Amico R, Deeks J J, Devereaux P J, Dickersin K, Egger M, Ernst E, Gøtzsche P C, Grimshaw J, Guyatt G, Higgins J, Ioannidis J P A, Kleijnen J, Lang T, Magrini N, McNamee D, Moja L, Mulrow C, Napoli M, Oxman A, Pham B, Rennie D, Sampson M, Schulz K F, Shekelle P G, Tovey D and Tugwell P 2009 Preferred reporting items for systematic reviews and meta-analyses: The PRISMA statement PLoS Med. 6 e1000097

Völker S and Kistemann T 2011 The impact of blue space on human health and well-being - salutogenetic health effects of inland surface waters: a review Int. J. Hyg. Environ. Health 214 449–60

White M P, Elliott L R, Gascon M, Roberts B and Fleming L E 2020 Blue space, health and well-being: A narrative overview and synthesis of potential benefits Environ. Res. 191 110169 Online: https://doi.org/10.1016/j.envres.2020.110169

Author Response

Dear Reviewer,

We sincerely thank reviewer for the valuable comments, which are of great help in revising the manuscript. Accordingly, the manuscriptrevised with comments has been systematically improved. Our responses are following. The page and line number mentioned below are in accordance with the number in revised manuscript.

Response to Comments:

Page 2, L. 4-5: This section is a bit out of place and, to me, does not follow the logic of the paragraph.

We thank reviewer for pointing this out. It has beenrevised as 'Many evidences have showed that natural spaces, especially urban blue and green spaces, can lower exposure to air pollution (extreme heat and noise), increase opportunities for physical activity and social engagement, and decrease stress and depression by contacting with nature (Barton, & Pretty, 2010; Coon et al., 2011)'. (Page 1; L14-17)

Page 2, L. 9-10: I am missing a reference to this statement or a least an examples of review studies in the area of toxicology and microbiology. In addition, I am not sure whether research on blue space and human health has been so much neglected, since there have been several reviews published on this topic (Gascon et al 2015, Völker and Kistemann 2011, Foley and Kistemann 2015, White et al 2020, McDougall et al 2020).

We thank reviewer for pointing this out. It has been revised as 'However, the literature about the relationship between water and human health was more assessed in environmental toxicology and microbiology in the past, and it mainly focused on the topic about drinking water security (Kumar, &Xagoraraki, 2010; Ni et al., 2010). Recently, the attention has more begun to turn to blue space and its benefits on human well-being (Domegan et al., 2016; Grellier et al., 2017)'. (Page 2; L13-17)

Page 2, L. 41-50: This Methodology of the review is very short and does not explain in detail how the final list of 85 articles was selected. I would recommend that the authors use the PRISMA guidelines or any other established guideline for systematic reviews (Moher et al 2009).

We thank reviewer for pointing this out. We have added Table 1 to supplement the collection, classification and analysis of the literature reviewing in this article. (Page 3)

Page 3, Figure 1: This figure seems a bit out a place. It nicely illustrates the human mental health recovery process after a disturbance, but the link to the health benefits of blue space is not very clear. Maybe the authors could add an explanation to the figure and its description.

We thank reviewer for pointing this out.We have emphasized 'blue space' as an external factor in the figure and figure captions.(Page 4)

Page 3, L. 9-10: The second objective is quite specific for a general literature overview on the design of natural and artificial blue space and blue space access. Could the authors please explain why you focus on the city of Suzhou. In general, could you please add this particular research aim to the introduction?

We thank reviewer for pointing this out. It has been revised as '(2) to propose the conceptual design strategies of building 'therapeutic landscape'. (Page 4; L2-3)

Page 4 Figure 2: Could the author please explain the novel aspects of their framework. In my opinion, the proposed framework is very similar to the framework published by (White et al 2020) and does not have any noticeable new aspects.

We thank reviewer for pointing this out. The framework shows the mechanisms of psychological and mental health promotion of natural environment and exercising in blue space. And we have added the section of design principles.(Page 5)

Page 5 L. 39-41 This sentence is hard to understand. Could you please rephrase it?

We thank reviewer for pointing this out. It has been revised as 'So that, in addition to the two established aspects of blue environments which are water features and accessibility, biodiversity also makes significant contributions to psychological health and is inextricably linked to psychological health (Campbell et al., 2011; Taylor, & Hochuli, 2015)'. (Page 7; L6-10)

Page 5 L. 42-43 The study by Cracknell et al. (2016) investigates the effect of marine biodiversity on people’s psychological well-being in a marine aquarium setting. I would suggest to add this information to the paragraph and discuss the possible implications, since experienced biodiversity in zoos or aquariums is not comparable to experienced biodiversity in nature.

We thank reviewer for pointing this out. It has been revised as 'And Cracknell et al. (2017)  took a study among 79 participants in UK and investigated the effect of marine biodiversity on psychological well-being in marine aquarium setting, and found that higher biodiversity would result in better psychological health. In contrast, evidences also indicated that the benefits of real nature will be even greater than simulated nature through making sure whether simulated environment can represent real environment (Kjellgren, &Buhrkall, 2010)'. (Page 7; L1-6)

Page L. 51: Could you please add information on the “respondents”. Are they part of the study conducted by Bell et al. (2015)?

We thank reviewer for pointing this out. It has been revised as 'Additionally, several respondents who live within four neighborhoods in the two coastal towns mentioned that their preference for three types of social dynamics, each of which was evident in blue space: (a) seeking friendly conversation and a joyful atmosphere, (b) engaging in spaces that offering a variety of opportunities for family leisure and well-being, and (c) bonding relationship through shared hobbies and experiences (Bell et al., 2015)'. (Page 7; L17-22)

Page 6 L 49: Please write “relaxing walks” instead of just “relaxing”.

We thank reviewer for pointing this out. It has been revised as 'Though these high-fence walls block the negative views of the river, they also block the opportunities for residents to take relaxing walks along the river and cut off the communication between residents and the river'. (Page 8; L13-15)

Page 7 L. 10: “As discussed above, …”

We thank reviewer for pointing this out.It has beenrevised as 'As above discussion, blue space promotes psychological health through (a) getting in touch with nature (e.g. visual contacting with animals and sound listening  from nature), (b) providing spaces for exercising (e.g. walking, talking and relaxing), and (c) creating biodiversity-rich sites (Grinde&Patil 2009). To meet these three main pathways by following design strategies, the site conceptual design will focus on (a) improving accessibility, (b) increasing interactivity and (c) restoring water quality and biodiversity'. (Page 8; L26-31)

Reviewer 2 Report

First of all, I would like to congratulate the authors for their beautiful and valid project, and for the effort to give it academic resonance through a scientific publication. However, regrettably, I cannot recommend its publication, as it lacks scientific rigor.

This article is initially presented as a exhaustive review article, and it is in this sense that the abstract presented is focused. However, the review is not exhaustive, the criteria for obtaining sources and rejecting other sources are not specified, it does not present a search time frame, nor does it clearly show the results of its review. It simply presents a theoretical exposition with various sources in no particular order, so there is no valid scientific process.

On the other hand, a new objective appears in the method that had not appeared until then (it does not appear in the abstract or in the introduction) which is the Panmen Inner-Town River project, this project has nothing to do with the review of literature and its link with it is unknown.

Finally, there is no discussion of results and the conclusions are presented ignoring the Panmen Inner-Town River project, so the final impression is that this paper is a mix of an intervention project merged with a review paper that is not directly linked and, therefore, it cannot be published in this state and its modification is so profound that its rejection and a new restart of the project are recommended.

My recommendation to the authors is that they separate both projects and present to an academic journal a complete and adequate theoretical review paper, on the one hand, and on the other hand, that they present the Panmen Inner-Town River project to another academic journal with all its parts well completed.

Author Response

Dear Reviewer,

We sincerely thank reviewer for the valuable comments.

First, we have set Table 1 in page 3 to discuss the classification and main ideas of the literature we reviewed. The literature can be divided into two categories: (a) the therapeutic benefits of blue space on psychological health, and (b) the positive influence of exercising on psychological health.

And on the suggestion of 'splitting the article into theoretical and case study parts', we have changed this article into a theoretical one, and develop the design model into a conceptual design model to make the article more complete and convincing. We removed the case example.

Round 2

Reviewer 1 Report

Review “Blue Space for Promoting Mental Health”

I thank the authors for addressing all of my comments from the previous review. The changes you made were able to improve the manuscript, especially in regard to the structure and readability of the manuscript. However, there are still two major issues that I am concerned with. And even though I mentioned both of them in the previous review, they were, in my opinion, not adequately addressed. I recommend to reject this manuscript, since this article does not meet the necessary standards of a novel scientific literature review.

Major issues:

Methods:

  • The literature review was not conducted according to any recommended scientific review guideline (e.g. PRISMA). Therefore, the article is not able to meet the scientific standard for a literature review.

Novelty:

  • Over the past years, several review studies have focused on the benefits of blue space. Therefore, most of the content and conclusions of this new article do not seem very new. The changes made by the authors after the first revision highlight blue space design principles as the key content. But since the case study has been removed and there is no discussion on different design approaches, this interesting aspect of the article lacks depth.

Further comments:

  1. 2, L. 26-27: The sentence does not make sense to me. Especially the part “… and emotional will response immediately after walking in blue space ...”. What does that mean?
  2. 3. Table 1: Thank you for adding this Table to the manuscript. Unfortunately, this addition does not fix the methodological issues that I raised in the previous review.
  3. 3, L. 10: It should be “Publications”
  4. 4, Figure 1: I still think that this figure does not really fit into the manuscript. In addition, the changes made to the figure by the authors do not help to better understand the figure`s content or message.
  5. 6, Figure 2: In my first review, I questioned the novelty of the author`s framework, since it is very similar to the framework published by (White et al 2020). In response, the authors added a section the figure about the proposed blue space design principles. In my opinion, this framework still lacks novelty.
  6. 9, L. 4-6: Where and when did you conduct a site analyses of urban rivers? How did you come to the conclusion that “… urban rivers are mostly blocked off from urban riverfront neighborhoods …”? In Europe, for example, not all urban rivers are blocked off from residential areas.
  7. 9, L. 22: “As discussed above, …”

Reviewer 2 Report

Congratulations on your work. The paper has improved clarity and organization significantly. However, the description of the method is still insufficient. As I mentioned in the first review, it is necessary:

- Specify date range for source search.
- Specify how many sources were initially found.
- Specify what criteria were followed to use some sources over others and why they were discarded (source exclusion criteria).
- It is also necessary a diagram that specifies the process followed throughout the investigation. That is, a diagram of the information flow through the different phases followed in the systematic review.
- It would also be advisable, although not mandatory, to specify the number of citations of the selected works to justify the adequacy of the criteria followed to select the sources.

Courage with the work to be done. Receive a cordial greeting.